# Effects of Phenolic Compounds on Walnut Bacterial Blight in the Green Husk of Hungarian-Bred Cultivars

**DOI:** 10.3390/plants11212996

**Published:** 2022-11-07

**Authors:** Géza Bujdosó, Éva Lengyel-Kónya, Mária Berki, Anita Végh, Attila Fodor, Nóra Adányi

**Affiliations:** 1Research Centre for Fruit Growing, Hungarian University of Agriculture and Life Sciences, 1223 Budapest, Hungary; 2Food Science Research Group, Institute of Food Science and Technology, Hungarian University of Agriculture and Life Sciences, 1118 Budapest, Hungary; 3Institute of Plant Protection, Buda Campus, Hungarian University of Agriculture and Life Sciences, 1118 Budapest, Hungary

**Keywords:** phenolic compounds, Hungary, nut development, state-approved cultivars, walnut blight immunity test, walnut phenology

## Abstract

The Persian walnut (*Juglans regia* L.) is the most grown nut tree crop in Central Europe. The aim was to study the full Hungarian walnut assortment with a distinct early spring phenology to detect the difference in phenolic profile in their green husks. Furthermore, the relationship between the presence and concentration of phenolic compounds and the tolerance/resistance of the observed cultivars to walnut bacterial blight was investigated. Examining the samples, significant differences were found between the concentrations of the different groups of phenolic compounds. Walnut blight immunity tests were also performed to clarify the role of phenolic compounds in the nut derived from a non-irrigated orchard. The Hungarian-bred local cultivars contained phenolic compounds in higher concentrations than the domesticated ones. There was a significant correlation between the budburst, as well as the pistillate flowers’ receptivity and the concentration of juglone. Cultivars with a low concentration of phenolic compounds were the most susceptible to walnut bacterial blight, except ‘Bonifác’.

## 1. Introduction

The Persian walnut (*Juglans regia* L.) is the most grown nut tree crop in Central Europe; its planting area and harvested yield are increasing annually in this region. All parts of the walnut such as leaves, green husk (mesocarp), pellicle, septum and kernel are rich in phytochemicals [1,2,3,4,5]. The nut is surrounded by the green husk that contains many bioactive compounds, such as polyphenols, which include flavonoids, [6,7], also known as phenolics [8,9,10], and pectin, glucans [5] and chlorophyll [11]. The phytochemicals have antimicrobial, antifungal and antioxidant activities [8,12,13,14,15,16,17,18,19].

The phenolic compounds play an important role in the plant defence mechanisms against stress caused by many environmental factors and various pathogens [20]. They can inhibit the growth of mycelia and different bacteria up to 50%, but it is indicated that their inhibition effects depend on their concentration and the relationship between the juglone and phenolics present in the plant tissue [8,12,13,14,15,16,17,18].

The phenolic content and its profile vary in all walnut plant organs [21,22]. The quantity of the compounds depends on the soil and climatic conditions [5,23], while the cultivar [8,10,24,25,26,27], the sample period inside the growing season [28], the use of extracting solvent [18,29,30,31,32] and cultivation technology all affect the phenolic content.

In the green husk of walnut, one of the most important bioactive compounds from a chemical point of view is phenolics. The main groups of phenolics are hydroxybenzoic acids, hydroxycinnamic acids, flavonoids and naphthoquinones, which were detected in green husk samples [27,28]. Dihydroxybenzoic acid, gallic acid and syringic acid together with their derivatives are hydroxybenzoic acids, while coumaric acid, chlorogenic acid, caffeic acid, ferulic acid and its derivatives or esters are classified in the group of hydroxycinnamic acids. The group of flavonoids includes myricetin, quercetin and naringenin and/or their derivatives, while naphthoquinones include juglone with its different derivatives, which is a characteristic compound of unripe husks of the nuts [27,28].

Unfortunately, the green husk is mostly an agricultural waste, but this valuable organ has some uses as traditional alcoholic drinks [33,34,35], bottled fruits, jams [36,37] and cold-pressed oil [38,39,40] and some cosmetics [41,42] can also be made from it.

The phenolic compounds also play an important role in the defence against walnut blight caused by *Xanthomonas arboricola* pv. *juglandis*. Gallic acid and juglone dominate in the nuts compared to the other phenolics [43]. The juglone concentration is usually higher in the green husk and nuts than in the leaves [44]. A negative correlation between juglone concentration and the susceptibility against walnut blight in the nuts was reported [45,46]. Juglone can be a discriminating biomarker for Xaj resistance [47,48] because the juglone concentration is smaller in the healthy areas than in the infected surfaces [49]. The resistance/tolerance of walnut cultivars against walnut bacterial blight does not depend on the density and breadth of stomas, the wax content of the leaves, leaf thickness, the thickness of the abaxial epidermis or the stratum corneum [50]. The inhibition effects of juglone depends on the isolation temperature too, as it has a negative correlation between antifungal activity and isolation temperature [18].

The aim of this study was to examine the full Hungarian walnut assortment, having a distinct early spring phenology, to detect the difference in phenolic profiles in their green husks. Furthermore, the relationship between the presence and concentration of phenolic compounds and tolerance/resistance of the observed cultivars to walnut bacterial blight at the stage of (finishing) completing nut size growth was investigated.

## 2. Results

### Phenolic Compounds

In the green husk of walnuts one of the most important bioactive compounds is the phenolics. The studied cultivars were examined to detect the difference in concentration of total phenolics and the various compounds. The total phenolic content of the green husk in the samples ranged between 4420 and 5740 mg GAE/100 g dry matter (d.m.). A series of studies looked at the total phenolic content of the green husk, usually examining the samples in the context of antioxidant capacity. The present results are in line with data published by Soto-Madrid and co-workers [7] where the range of the total phenolic compounds was between 3117 and 10,601 mg GAE/100 g d.m. The research group of Soto-Maldonado [42] obtained the total polyphenol content as 1862.9 ± 72.4 mg GAE/100 g d.m., measured in the tested sample, while Oliveira and co-workers [8] determined the total phenolic content of the green husk of different cultivars and found that their concentration ranged from 32.61 mg/g of GAE (cv. ‘Mellanaise’) to 74.08 mg/g of GAE (cv. ‘Franquette’). Despite many applications, the Folin–Ciocalteu method [51] is only suitable for a rough estimate because, as Scalbert and co. [52] described, the molar absorbance or, instead, molar absorptivity per reactive group obtained for different compounds varies; in other words, the color generated by the different phenolic compounds depends on the molecular structure.

There is limited information in the scientific literature on detailed analysis of the walnut green husk; however, it can be important when we take into account the timings of the different developmental stages of the examined cultivars. When examining the cultivars, the concentration difference of the various compounds was also determined. Figure 1 and Figure 2 represent the chromatograms of the ‘Bonifác’ sample measured on a 280 nm wavelength for the main phenolic components and on 420 nm for juglone determination.

Examining the samples, significant differences were found between the concentrations of the different groups of phenolic compounds. In our measurements all components were calculated in rutin equivalent, except for the naphthoquinones calculated in the juglone equivalent. Figure 3 depicts the phenolic compound composition of the samples. The green husk of the ‘Alsószentiváni’-type samples contained the highest concentration of the components in question, especially ‘Bonifác’, in which the concentrations of hydroxybenzoic acids, hydroxycinnamic acids, flavonoids and naphthoquinones were 3176 mg/100 g d.m., 1947 mg/100 g d.m., 2203 mg/100 g d.m. and 1275 mg/100 g d.m., respectively. Comparing these results to the control ‘Chandler’, the total content of phenolics in the green husk was about 3.7 times higher in ‘Bonifác’. The total phenolics varied in the samples between 2329 and 8601 mg/100 g d.m.

The concentration of the hydroxybenzoic acids varied between 407 and 3176 mg/100 g d.m., with the samples with higher values being ‘Alsószentiváni 117’, ‘Alsószentiváni kései’, ‘Bonifác’ and ‘Köpcös’, 1220 mg/100 g d.m., 2063 mg/100 g d.m., 3176 mg/100 g d.m. and 1283 mg/100 g d.m., respectively. ‘Milotai 10′ and ‘Milotai kései’ contained 802 mg/100 g d.m. and 764 mg/100 g d.m., while ‘Milotai intenzív’ and ‘Chandler’ had the lowest values, 423 mg/100 g d.m. and 407 mg/100 g d.m., respectively. The content of hydroxycinnamic acids did not show such great differences; the concentration varied between 723 and 1947 mg/100 g d.m. In terms of flavonoids, a similar trend was seen; the concentration varied between 741 mg/100 g d.m. and 2203 mg/100 g d.m. The content of naphthoquinones was the highest in ‘Bonifác’ (1275 mg/100 g d.m.) and ‘Milotai kései’ (1260 mg/100 g d.m.). The lowest concentrations of this compound group were detected in ‘Köpcös’ (499 mg/100 g d.m.) and ‘Chandler’ (458 mg/100 g d.m.) (Figure 3).

## 3. Discussion

### 3.1. Phenolic Compounds and Early Spring Phenology

A higher concentration of phenolic compounds was detected in this trial compared to the data found in the literature [8,9,10,24,25,26,27,53]; whereas the mentioned authors collected their samples before ripening, our samples were collected during the lignification period in late June, when these components’ amounts reach their peak [28,51,53]. The juglone concentration measured in this trial was similar to the results of the research group from Slovenia [54]. This literature source also confirms that the locally bred cultivars can produce a higher concentration of compounds compared to the domesticated ones. 

For juglone, several authors reported data that are consistent with our results. A research group [42] reported the amount of juglone was 169.1 mg/100 g d.m. and another research group [53] published values between 218 and 1404 mg/100 g d.m. depending on sampling time, while some researchers from Romania [10] found measurements between 20.56 and 42.78 mg/100 g from different varieties of mature walnut green husk. These results are in the same order of magnitude to ours, but, at the same time, they indicate the great variation due to the sampling time. 

The ‘Alsószentiváni 117′, ‘Milotai 10′, ‘Tiszacsécsi 83′ and ‘Köpcös’ cultivars started their budburst and pistillate flowers’ receptivity almost at the same time as the control cultivar ‘Chandler’. The budburst of ‘Milotai intenzív’ was similar to the control, but it needed the longest period among the observed cultivars for the pistillate flowers to appear, 7 to 12 days after budburst. The budburst and pistillate flowers’ receptivity of ‘Bonifác’, ‘Milotai kései’ and ‘Alsószentiváni kései’ were 3 to 12 days and 7 to 17 days later than the control, respectively. 

Furthermore, there were strong significant correlations between the budburst, as well as the pistillate flowers’ receptivity, and the concentration of juglone, where R^2^ was 0.80 and 0.92, respectively (Figure 4.). The lowest values were measured in samples that were the earliest in their pistillate flowers’ receptivity, namely, ‘Alsószentiváni 117’ (192 mg/100 g d.m.), ‘Milotai 10’ (250 mg/100 g d.m.), ‘Tiszacsécsi 83’ (170 mg/100 g d.m.), ‘Köpcös’ (160 mg/100 g d.m.) and ‘Chandler’ (160 mg/100 g d.m.). (Figure 3.) The ‘Alsószentiváni kései’, ‘Bonifác’, ‘Milotai intenzív’ and ‘Milotai kései’ cultivars reached a similar state of development about 10–15 days later, and the juglone content was measured much higher (512 mg/100 g d.m., 568 mg/100 g d.m., 345 mg/100 g d.m. and 714 mg/100 g d.m., respectively) compared to those varieties, which had earlier budburst and blossom (Figure 3.). 

Discriminant analysis was performed to determine if the different cultivars could be distinguished according to the grouping variables, such as budburst and pistillate flowers’ receptivity based on the given analytical tests. The sums of the four data groups and juglone concentration were used as independents (Figure 5 and Figure 6). Based on the results, the date of the budburst is less indicative than the pistillate flowers’ receptivity. When analyzing the results, the discriminant analysis could not distinguish between ‘Milotai 10′, ‘Milotai intenzív’, ‘Köpcös’ and ‘Chandler’, as they were very similar in the time of budburst (107–109 calendar days) but differed in analytical values. When using all five data series for discriminant analysis, 89% of the original grouped cases were correctly classified. In contrast, when the data of pistillate flowers’ receptivity was used for the discriminant analysis, it was found that the amount of flavonoids, naphthoquinones and juglone as variables already ensured the correct classification of 100% of the original grouped cases.

Table 1 contains the correlations between early spring phenology and the observed phenolic compounds. The naphthoquinones correlated well to the budburst (0.70) and the blossom time (0.78). Interestingly there was a strong correlation between the budburst and the blossom of the pistillate flowers (0.79), as without a shoot, it is not possible for the female flower to develop. 

### 3.2. Tolerance/Susceptivity to Xanthomonas Arboricola pv. Juglandis

The dendrogram, containing the diameter of the necrotic spots and the disease rate values, differentiated the cultivars belonging to certain susceptibility groups. In 2020, the susceptibility/resistance of cultivars showed significant differences. Based on the statistical evaluation of the data, ‘Milotai intenzív’ proved to have a high susceptibility (hS), while susceptibility (S) was detected for ‘Bonifác’, ‘Tiszacsécsi 83’, ‘Alsószentiváni kései’, ‘Milotai 10’ and ‘Chandler’. Moderately susceptible cultivars were ‘Milotai kései’ and ‘Alsószentiváni 117’. The origin of the Hungarian walnut cultivars did not relate with their susceptibility to walnut blight (Figure 7).

In previous research a linkage between the polyphenols content and resistance to walnut bacterial blight was determined [28]. A strong correlation between the high polyphenols content and resistance to walnut blight was observed in this study. The current results confirmed this statement, except in the case of cultivar ‘Bonifác’. This cultivar contained the highest concentration of phenolic compounds, but its nuts were susceptible to walnut blight (Figure 7) as described in the cultivars’ description [55]. In addition to the phenolic compounds, there are more factors, e.g., protein profile of the green husk [56], to determine the susceptibility of a genotype against walnut blight. Little research about the susceptivity of ‘Alsószentivani 117′ and its hybrids to walnut blight is known, and further research needs to investigate this. Based on the artificial infections, the varieties with late budburst usually had better resistance to walnut blight due to them having a higher amount of phenolic compounds (Figure 3 and Figure 7). However, a high concentration of phenolic compounds does not only link to resistance. Some cultivars with an early budburst can also have good resistance, such as ’BD6’ [57]. This conclusion confirms other research results [45,46,47,48,54] that there is a negative correlation between juglone concentration and the susceptibility of the walnut genotypes and cultivars to blight. 

The concentration and composition of phenolic compounds in the green husk of walnuts change significantly during crop development; thus a detailed kinetic study would be very important.

Evaluating the results, it is concluded that the concentration of phenolic compounds of the most susceptible cultivars, ‘Milotai intenzív’ ‘Tiszacsécsi 83’ and ‘Chandler’, was the lowest during the measurements. 

## 4. Materials and Methods

### 4.1. Plant Material

The trial was conducted at the Experimental Fields of the Hungarian University of Agriculture and Life Sciences Research Centre for Fruit Growing (GPS coordinates: 47°20′11.44″ N 18°51′53.42″ E). The trial was planted in spring 1990 on chernozem soil with high lime (pH = 8, total lime content in the top 60 cm layer 5%) and humus content (2.3–2.5%). Considering the Arany-type cohesion index [58] the K_A_ = 40 refers to medium compactness. Meteorological conditions of the site are presented in Table 2. 

The entire Hungarian walnut assortment (‘Alsószentiváni 117′, ‘Milotai 10′, ‘Tiszacsécsi 83′ selected from the local Hungarian population; ‘Milotai intenzív’, ‘Milotai kései**℗**’(kései refers to its late leafing time) crossbred between ‘Milotai 10’ and ‘Pedro’; ‘Alsószentiváni kései**℗**’, ‘Bonifác**℗**’ crossbred between ‘Alsószentiváni 117’ and ‘Pedro’), found on the Hungarian National Variety List, was established in the trial. All are state-registered cultivars for Hungarian and neighboring countries’ (e.g., Slovakia, Romania, Serbia, Croatia, Slovenia and Austria) production. Another genotype (possible candidate) for the rootstock ‘Köpcös’ was added to the trial. The control of this trial was the US-bred ‘Chandler’, with detailed phenological descriptions in Table 3. 

The trees were grafted on selected *Juglans regia* seedlings, and planted 10 × 10 m in the row and between the rows. The grafted trees were trained as a central leader canopy, with 5 trees replicated of each, and the trial was not irrigated. 

### 4.2. Phenological Stages

All the phenological stages were recorded as indicated in the Ctifl scheme [59]. The corresponding stages when the data was collected, were the budburst stage of ‘Cf’; the start of pistillate flowers’ receptivity, the stage of ‘Ff2’. Thus the start of pistillate receptivity period was considered when the stigmas in the earliest pistillate flowers became receptive onwards. The data were recorded within two to three day intervals, usually in the morning. 

### 4.3. Sample Collection and Preparation

The samples were collected late June, on the 181st day of 2020, during lignification of the nut shell. The time of budburst and pistillate flowers’ receptivity in days of each samples is summarized in Table 4. During sample collection, the following periods were taken from the pistillate flowers’ receptivity until the sample collection: for ‘Alsószentiváni 117’, ‘Milotai 10’, ‘Tiszacsécsi 83’, ‘Köpcös’ 63 days, for ‘Chandler’ 62 days, for Milotai intenzív’ 53 days, for ‘Alsószentiváni kései’ and ‘Bonifác’ 48 days, as well as 45 days for ‘Milotai kései’. The collected nuts were immediately delivered to the laboratory where ten nuts of every sample type sample were peeled with a 2 mm width blade kitchen peeler and the green husk was lyophilized. The dried samples were ground to fine powder and stored hermetically sealed at 4 °C until analysis.

### 4.4. Chemicals

Standards used for identification and quantification of phenolic compounds were gallic acid, syringic acid, catechin, chlorogenic acid, caffeic acid, p-coumaric acid, quercetin, rutin (quercetin-3-O-rutinoside), hyperoside (quercetin-3-O-galactoside), kaempherol, juglone (Merck, Darmstadt, Germany), ferulic acid and ellagic acid (Fluka-Honeywell, Charlotte, NC, USA). HPLC/MS grade acetonitrile and formic acid, analytical grade methanol and acetic acid were purchased from Avantor (Radnor, PA, USA). The bidistilled water was filtered through a 0.45 µm nylon membrane.

### 4.5. Total Phenolic Content

Total phenolic content was determined by the Folin–Ciocalteu’s photometric method [53]. Fifty mg of green husk was extracted in 10 mL of 80% methanol, left for 24 h. After 30 min shaking at 25 °C the mixture was filtered. One hundred µL of the filtrate was mixed with 3.7 mL distilled water, 0.5 mL Folin–Ciocalteu reagent and 2 mL 20% Na_2_CO_3_ solution and filled up to 10 mL with distilled water. Samples were left in the dark for 30 min at room temperature; the absorbance was measured by spectrophotometer (UY-160 A, Shimadzu, Kyoto, Japan) at 750 nm against blank sample (4.75 mL distilled water and 0.25 mL Folin–Ciocalteu reagent). Total phenolic content was calculated according to a calibration curve prepared with gallic acid in 0–0.5 mg/mL concentration range (Sigma-Aldrich, St. Louis, MO, USA). 

### 4.6. Phenolic Compounds’ Determination by HPLC-ESI-DAD

One hundred mg of dried green husks were placed in a test tube and 10 mL of the extraction solvent (bidistilled water/2% acetic acid in methanol, 30/70) added. Mixtures were sonicated (AU 65, ArgoLab, Carpi, Italy) for 20 min at room temperature. After centrifugation (5000 rpm, 20 min; Digicen 21, Orto Alresa, Madrid, Spain), the supernatant was filtered through a 0.45 µm PVDF syringe filter before HPLC injection and analysed on the same day. 

A Waters Alliance system (Waters, Milford, (CT), USA) consisting of a Model e2695 separation module with Model 2998 photodiode array (PDA) detector operated by Empower software (Waters, Milford, (CT), USA) was used for the determination of the phenolic compounds. The separation was carried out using a Sphinx 5 μm 250 × 4.6 mm column (Macherey-Nagel, Düren, Germany) by gradient elution. The mobile phase was (A) 0.1% formic acid in bidistilled water and (B) 0.1% formic acid in acetonitrile. The gradient elution program: 0–9 min, 5–20% B; 9–16 min, 20–22% B; 16–25 min, 22–50% B; 25–28 min 50% B; 28–40 min, 50–100% B; 40–43 min, 100% B; 43–45 min, 100–5% B; 45–50 min, 5% B. The injected volume was 20 µL and the flow rate was 0.7 mL/min. Data acquisition of PDA proceeded with the range of 200–600 nm and the detection wavelengths were at 280, 320, 355 and 420 nm.

The HPLC system was coupled to a Model Acquity Mass (QDa) detector (Waters, Milford, (CT), USA). The mass spectrometry conditions were set as both negative and positive modes: electrospray ionization (ESI) was used as a source, mass spectra in the *m*/*z* range from 100 to 1000 were obtained and probe temperature was adjusted to 600 °C (default). In positive ion mode the cone voltage was set to 15 V and capillary voltage was 1.5 kV. In negative ion mode those were 50 V and 0.8 kV, respectively. 

Identification of phenolic compounds was achieved by comparing their spectral characteristic, retention times, measured mass (*m*/*z*) and fragmentation pattern. In addition, the previously published literature [26,60,61,62] and internet databases [63,64,65] were applied. The quantification of the identified phenolic compounds was based on calibration curve of rutin at 280, 320 and 355 nm as rutin gives an easily measurable signal at these wavelengths, and juglone at 420 nm.

### 4.7. Walnut Blight Immunity Test

The susceptibility test was carried out based on the methods [66,67] using 10 immature green nuts from every individual, collected in 2020. For the artificial infection a mixture of 3 Xaj strains was used, isolated from naturally infected walnut nuts from four different locations in Hungary (Vecsés, Karád, Balatonbolgár and Zánka). Before infection of immature nuts, isolates were either inoculated into the intercellular tissue of tobacco leaf (‘White Burley’) or were inoculated in immature nuts for confirmation capability of them to induce hypersensitive tissue necrosis and also aggressiveness to produce disease symptoms. The mixture of the three strains was used for the infection after the virulence test. Before the infection the fruit surface was disinfected with alcohol. Two inoculations of 20 μL bacterial suspension were performed in exocarp for each nut, thus the 10 nuts per cultivar with 2 inoculations each were tested. Sterile distilled water (SDW) was injected into the 10 immature nuts for every cultivar as control negative treatment. After the infection, the nuts were incubated in transparent plastic boxes for 7 days at a temperature of 26–28 °C, with over 90–95% relative humidity. Temperature and RH% were monitored by a micro-sensor placed into one of the plastic boxes. The disease severity was recorded on a scale from 0 to 4 (Figure 8), from the least to the most, based on the diameter and depth of necrosis reached on the 7th day: 0—no symptoms; 1—less than 2.0 mm, superficial and small spots on the inoculation point; 2—blackening on the inoculation point of nut by 2.1 mm to 3 mm; 3—blackening on the inoculation point of nut by 3.1 mm to 4 mm; 4—blackening on the inoculation point of nut more than 4.1 mm; see legend of Figure 8.

Four groups were formed: Moderately Resistant (MR) ≤ 2 mm; Moderately Susceptible (MS) 2.1–3 mm; Susceptible (S) 3.1–4 mm; Highly Susceptible (HS) ≤ 4.1 mm. The test was performed using the same cultivars as for chemical analysis except for ‘Köpcös’.

### 4.8. Statistical Analysis

The data derived from compositional analyses and the immunity test were evaluated using the SPSS software (IBM SPSS 27.0, Chicago, IL, USA). Discriminant analysis was carried out based on the results of analytical measurements. Relationships between the observed traits in the correlation matrix were calculated by Pearson correlation coefficient. For the immunity test the statistical analyses were determined based on the sample size, distribution analysis (Kolmogorov–Smirnov test) and *t*-test. It was accepted on the *p* ≤ 95% confidence level. A hierarchical cluster analysis based on one-year data regarding the diameter and disease rate was conducted in order to classify the cultivars into susceptibility groups. The results were represented in a dendrogram (Figure 7). Values represent the mean and standard deviation of three replicates from each sample.

## 5. Conclusions

The research groups from Slovenia [28,54] indicated in their study that the concentration of phenolics in the walnut crop varies with time inside the growing season. It can be assumed that the concentration and composition of phenolic compounds in the green husk of walnuts also change significantly during crop development; thus, a detailed kinetic study would be very important in the future [54].

The concentration of phenolic compounds varied by cultivars [8,10,24,25,26,27]; it was higher in the locally-bred cultivars compared to the foreign-bred ‘Chandler’, grown and collected among Hungarian climatic conditions. The phenolic compounds inhibited the artificial *Xanthomonas arboricola* pv. *juglandis* infection on the surface of the green husk. This effect had a strong correlation with the high phenolics’ concentration, except on ‘Bonifác’. 

Naphthoquinones showed strong correlations between budburst and blossom time, which are related to their yearly fluctuation; the earlier the budburst and blossom, the earlier their peak concentration can be reached [54]. The cultivars with late budburst and blossom time are the best for growers due to their lower susceptibility to walnut blight as described in some previous papers [68,69,70]. Fortunately, there is a strong correlation between the budburst and blossom time (0.79) too, as described by breeders [68,69,70].

## Figures and Tables

**Figure 1 plants-11-02996-f001:**
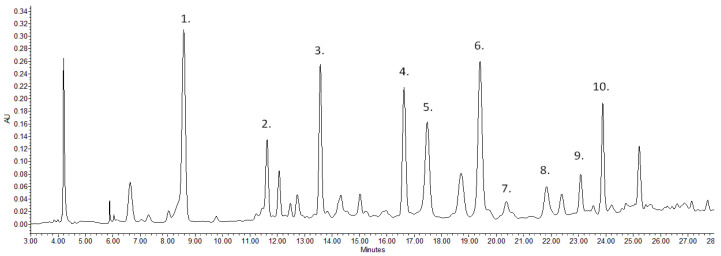
Chromatogram of ‘Bonifác’ sample for the determination of the main identified phenolic components (λ = 280 nm; (1) gallic acid, (2) neochlorogenic acid, (3) p-coumaric acid, (4) syringic acid, (5) methylmirecitin derivative 1, (6) trihydroxyisoflavanone derivative, (7) quercetin-hexoside; (8) methylmirecitin derivative 2, (9) methoxynaringenin derivative, (10) quercetin-pentoside).

**Figure 2 plants-11-02996-f002:**
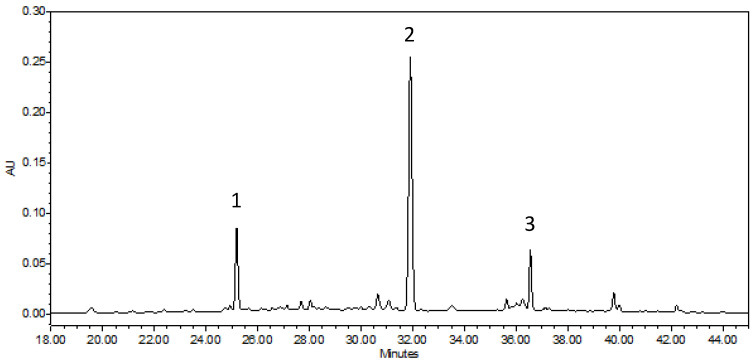
Chromatogram of ‘Bonifác’ sample for the determination of juglone (λ = 420 nm).

**Figure 3 plants-11-02996-f003:**
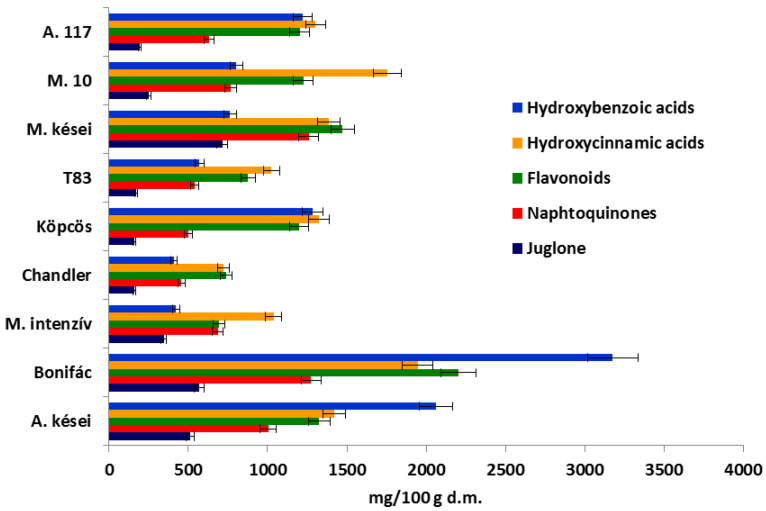
Composition of phenolic compounds measured in the green husk of walnut samples (A.: Alsószentiváni, M.: Milotai).

**Figure 4 plants-11-02996-f004:**
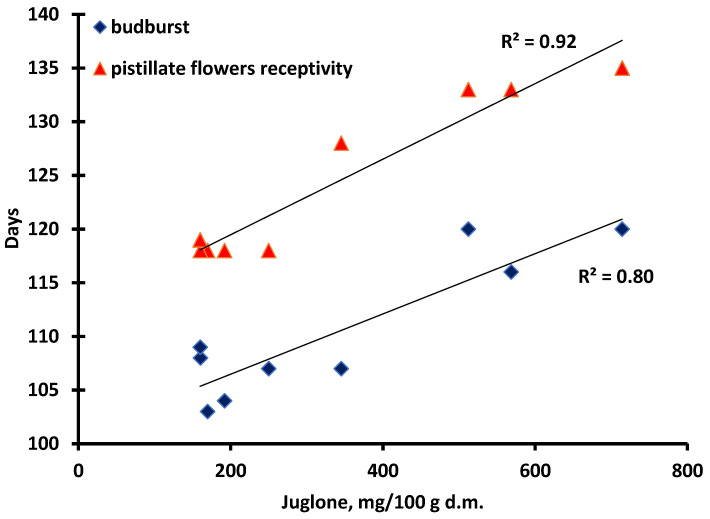
Correlation between the budburst and pistillate flowers’ receptivity and the concentration of juglone.

**Figure 5 plants-11-02996-f005:**
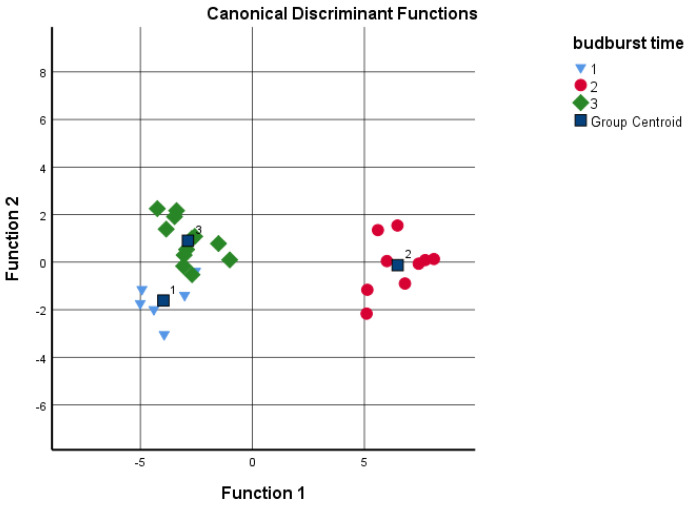
Discriminant analysis by budburst time as a grouping variable; 1—early: 103–104 days; 2—medium: 107–109 days; late: 116–120 days (independents: sum of hydroxybenzoic acids, hydroxycinnamic acids, flavonoids and naphthoquinones, as well as juglone concentration).

**Figure 6 plants-11-02996-f006:**
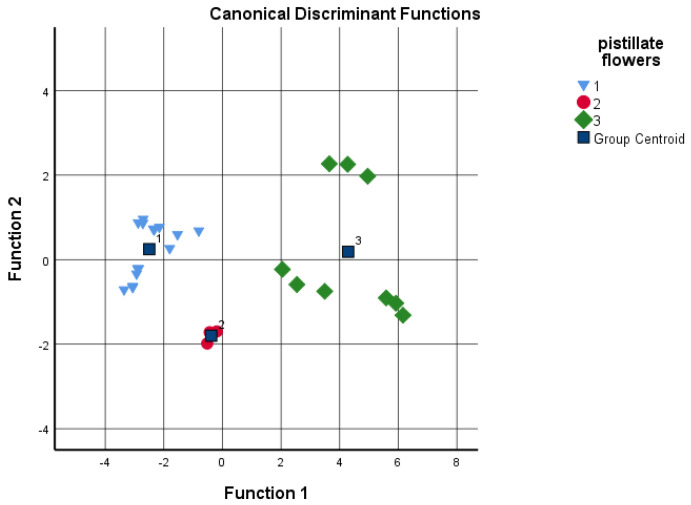
Discriminant analysis by pistillate flowers’ receptivity time as a grouping variable; 1-early: 118–119 days; 2-medium: 128 days; late: 133–135 days (independents: sum of flavonoids and naphthoquinones, as well as juglone concentration).

**Figure 7 plants-11-02996-f007:**
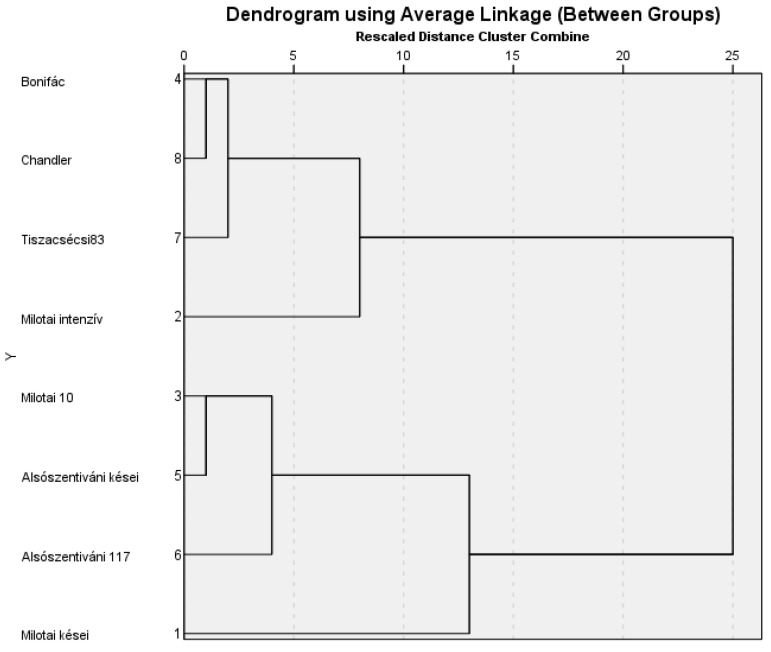
Dendrogram of Hungarian-bred walnut cultivars from the point of view of their susceptibility to walnut blight.

**Figure 8 plants-11-02996-f008:**
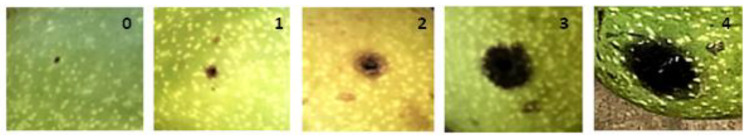
Evaluation of susceptibility, scale (0–4). 0—no symptoms; 1—less than 2.0 mm, superficial and small spots on the inoculation point; 2—blackening on the inoculation point of nut by 2.1 mm to 2.6 mm; 3—blackening on the inoculation point of nut by 2.7 mm to 3.1 mm; 4—blackening on the inoculation point of nut more than 4.1 mm.

**Table 1 plants-11-02996-t001:** Correlation matrix of measured parameters and observed characteristics (R^2^) *.

	Budburst	Blossom Time
hydroxybenzoic acids	0.27	0.27
hydroxycinnamic acids	0.15	0.14
flavonoids	0.37	0.31
naphthoquinones	0.70	0.78
total phenolic acids	0.40	0.35
budburst		0.79

*: Samples were collected during lignification of the nuts.

**Table 2 plants-11-02996-t002:** Meteorological data during the data collection 2020.

Parameters	Value
average yearly temperature	11.4 °C
average yearly temperature during the growing season (March–September)	16.1 °C
average yearly luminous flux	1015 L/m^2^/day
average yearly precipitation	434.1 mm
annual average of sunshine hours	2065

**Table 3 plants-11-02996-t003:** Detailed phenological descriptions of the examined walnut cultivars in this study [55].

Cultivars	Nut Size (Diameter) (mm)	Dried Kernel Weight (g/Nut)	Kernel Color
Alsószentiváni 117	33–36	4.4–6.2	medium brown
Milotai 10	33–35	5.2–7.5	light yellow
Tiszacsécsi 83	32–34	4.4–5.3	medium brown
Köpcös	32–34	4.6–4.8	medium brown
Chandler	30–32	4.2–4.5	light yellow
Milotai intenzív	32–34	4.4–5.3	light yellow
Bonifác	32–34	4.8–6.2	light brown
Milotai kései	32–34	5.3–6.6	light brown
Alsószentiváni kései	32–35	4.4–6.2	light brown

**Table 4 plants-11-02996-t004:** The time of budburst and pistillate flowers’ receptivity in calendar days starting from 1st January.

	Budburst (Days) ^1^	Pistillate Flowers’ Receptivity (Days) ^2^	Days after Receptivity before Sampling ^3^
Alsószentiváni 117	104 a	118 a	63 a
Milotai 10	107 ab	118 a	63 a
Tiszacsécsi 83	103 a	118 a	63 a
Köpcös	108 ab	118 a	63 a
Chandler	109 ab	119 a	62 a
Milotai intenzív	107 ab	128 b	53 b
Bonifác	116 b	133 c	48 c
Milotai kései	120 c	135 c	45 c
Alsószentiváni kései	120 c	133 c	48 c

SD (5%) ^1^ = 2.3, SD (5%) ^2^ = 3.1, SD (5%) ^3^ = 2.9.

## Data Availability

All data were collected from the published research papers.

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
