# Peer review of "Effects of Phenolic Compounds on Walnut Bacterial Blight in the Green Husk of Hungarian-Bred Cultivars"

_plants, 2022, doi:10.3390/plants11212996_

Round 1

Reviewer 1 Report

The few suggestions are attached.

Author Response

Dear Reviewer,

first of all I like to take the opportunity to say Thank you for your time and advises, which can improve the quality of our paper titled “Effects of phenolic compounds on walnut bacterial blight in the green husk of Hungarian bred cultivars”.

There you can find our answers:

Line 70 meaning of d.m. were written in full (Line 82 in the revise manuscript.)

Line 323 meaning of d.m. wwas written at the first mention in the main text.

Line 324 meaning of d.m. was written in full at the first mention, meaning of SDW is in the line 323 of the main text.

Line 547 the grey background was removed behind the doi number

Yours sincerely, 

the authors

Reviewer 2 Report

The paper "Effects of phenolic compounds on walnut bacterial blight in the green husk of Hungarian bred cultivars" deals with the evaluation of different phenol and polyphenol compositions in different Persian Walnut samples. The paper needs to be deeply revised before publication, starting from a general revision of the English language throughout the manuscript.

In general, the aim of the paper is unclear. The brief introduction section, which cites an enormous amount of previous papers without any discussion, does not provide any indication. 

A clearer description of the samples must be added. What are these 9 different nuts? 

"The 360 concentration of total phenol acids are correlated well to the concentration of the phenolic compounds, therefore we got strong correlations between total phenolic acids and hydroxybenzoic acid, hydroxycinnamic acids, flavonoids, and naphtoquinones." This sentence makes no sense, given that phenolic acids (and non phenol acids) are phenolic compounds and that hydroxybenzoic and hydroxycinnamic acids are phenolic acids. Why would it be a significant result that hydroxybenzoic/hydroxycinnamic acids are correlated to the content of phenolic acids?

Author Response

Dear Reviewer,

first of all I like to take the opportunity to thank you for your time and advises, which can improve the quality of our paper titled “Effects of phenolic compounds on walnut bacterial blight in the green husk of Hungarian bred cultivars”.

There you can find our answers:

English language of the main text was revised by a native speaker.

Aim of the paper was added and stated in the lines 71-75.

The introduction was revised and completed with some indications. There is not big debate in the literature sources on the field of phenolic compounds.

More detailed information about the examined cultivars was added to the second paragrapf of 4.1. Furthermore, short description of the samples was added in the Table 3.

Why would it be a significant results that hydroxybenzoic/hydroxycinnamic acids are correlated to the content of phenolic acids?

Based on your question the Table 1 was revised.

Table 1. Correlation matrix of measured parameters and observed characteristics (R2)*

bud-

burst

blossom time

hydroxybenzoic acids

0.27

0.27

hydroxycinnamic acids

0.15

0.14

flavonoids

0.37

0.31

naphtoquinones

0.70

0.78

total phenolic acids

0.40

0.35

budburst

0.79

*: Samples were collected during lignification of the nuts

Yours sincerely,

the authors

Reviewer 3 Report

Dear authors, 

The  introduction is rather basic and grossly generalised, please be more specific, try using scientific language. 

Overall English must be improved trhoughout the manuscript and especially the intorduction. 

What is the aim of the article?? --> must be stated!!!

As simple as Figure 3 is, it is not clear what the A, M stand afor and why are they not mentione din all the cv names.

What do you measure with R2? check the difference between R2 and R!

The discussion is all over the place. Also the performed experiments dont really justify the conlcusions- they are very basic. 

The quality should be improved for publication, at this stage this manuscript seems like a compilation of known facts un any concrete experiments backing it. 

Author Response

Dear Reviewer,

first of all I like to take the opportunity to thank you for your time and advises, which can improve the quality of our paper titled “Effects of phenolic compounds on walnut bacterial blight in the green husk of Hungarian bred cultivars”.

There you can find our answers:

Language of the manuscript (especially introduction) was improved by a native speaker.

Aim of the paper was added and stated in the lines 71-75.

The local bred cultivars have traditional, Hungarian names, which are a bit long. We checked the Fig. 3. and corrected the cultivar names.

R2 is the correlational co-efficient, the authors marked this value all times in the revised version.

The conclusions were revised.

Yours sincerely, 

the authors

Round 2

Reviewer 2 Report

The revisions on the manuscript do not fulfill the concerns expressed in the previous round of revision. Therefore, I regret to say that the manuscript is not suitable for publication.

Author Response

Dear Reviewer2, 

Thank you for your answer. 

Yours sincerely, 

the authors